# Exploring the multiple policy objectives for primary care networks: a qualitative interview study with national policy stakeholders

Kath Checkland,[1] Jonathan Hammond  ,[1] Lynsey Warwick-Giles,[1] Simon Bailey[2]

[1]Division of Population Health, Health Services Research, and Primary Care; School of Health Sciences; Faculty of Biology, Medicine and Health, The University of Manchester, Manchester, UK
[2]Centre for Health Services Studies, University of Kent, Canterbury, UK

**Correspondence to**
Dr Kath Checkland;
Katherine.H.Checkland@manchester.ac.uk

## ABSTRACT

**Objectives** English general practice is suffering a workforce crisis, with general practitioners retiring early and trainees reluctant to enter the profession. To address this, additional funding has been offered, but only through participation in collaborations known as primary care networks (PCNs). This study explored national policy objectives underpinning PCNs and the mechanisms expected to help achieve these, from the perspective of those driving the policy.

**Design** Qualitative semistructured interviews and policy document analysis.

**Setting and participants** National-level policy maker and stakeholder interviewees (n=16). Policy document analysis of the Network Contract Direct Enhanced Service draft service specifications.

**Analysis** Interviews were transcribed, coded and organised thematically according to policy objectives and mechanisms. Thematic data were organised into a matrix so prominent elements can be identified and emphasised accordingly. Themes were considered alongside objectives embedded in PCN draft service delivery requirements.

**Results** Three themes of policy objectives and associated mechanisms were identified: (1) supporting general practice, (2) place-based interorganisational collaboration and (3) primary care 'voice'. Interviewees emphasised and sequenced themes differently, suggesting meeting objectives for one was necessary to realise another. Interviewees most closely linked to primary care emphasised the importance of theme 1. The objectives embedded in draft service delivery requirements primarily emphasised theme 2.

**Conclusions** These policy objectives are not mutually exclusive but may imply different approaches to prioritising investment or necessitate more explicit temporal sequencing, with the stabilisation of a struggling primary care sector probably needing to occur before meaningful engagement with other community service providers can be achieved or a 'collective voice' is agreed. Multiple objectives create space for stakeholders to feel dissatisfied when implementation details do not match expectations, as the negative reaction to draft service delivery requirements illustrates. Our study offers policy makers suggestions about how confidence in the policy might be restored by crafting delivery requirements so all groups see opportunities to meet favoured objectives.

### Strengths and limitations of this study

► Primary care networks represent a significant policy development in England, and we offer the first systematic analysis of national policy objectives as articulated by a range of different stakeholders.

► We interviewed 16 national-level policy makers or stakeholders working in a range of organisations, including NHS England and NHS Improvement, and in government in the Department of Health and Social Care.

► National healthcare policy objectives are rarely subject to critical academic attention in the early stages of policy implementation, but doing so provides scope for better understanding challenges a policy may face and developing strategies to address these.

► This is a fast moving policy area and our results inevitably reflect a particular snapshot of time.

## INTRODUCTION

Primary care in the UK is in crisis. Young doctors are not entering, or remaining in, the specialty in sufficient numbers to cope with demand, and many older general practitioners (GPs) are increasingly dissatisfied with general practice due to various intrinsic and extrinsic factors and are choosing to retire early.[1 2] The problem is complex, arising from the intersection between demographic change, rising population expectations of what services should deliver, and funding growth which has not matched the rise in costs.[3 4] In keeping with this complexity, potential solutions are multiple, with for example the GP Forward View[5] offering a number of potential policy fixes. Most recently, consensus has gathered around the idea that GP practices—traditionally independent contractors to the National Health Service (NHS)—should work together more effectively to support one another and provide a broader range of services.[6] To enable this, a centrepiece of the new NHS Long Term Plan[7] is the creation and funding

of Primary Care Networks (PCNs). The underlying idea is relatively simple: incentivise GP practices to combine together into groups, so they can find economies of scale, employ a wider range of staff, link more effectively with community-based providers, and through these improve services to patients.

Collaborations between GP practices are not new, in the UK or elsewhere. Internationally a 'polyclinic' model is said to offer the advantages of person-centred primary care alongside an extended range of services from a multi-disciplinary team.[8–10] In the UK, practices have voluntarily formed themselves into collaborative groups to undertake collective audits,[11] deliver out-of-hours services[12] and support commissioning.[13] Outcomes from these previous collaborations have been mixed. The 'polyclinic' model struggled to establish itself in England,[14] but other collaborative ventures such as GP cooperatives were successful, particularly in improving GP job satisfaction.[15 16] A recent study of GP federations (voluntary collaborations between GP practices) found 'successful' federations were more likely to have coalesced around a particular service to be delivered or problem to be solved, and that collaborations took time to develop. Importantly, the study found that federations struggled with running costs and required considerable managerial expertise to support them.[17]

PCNs are an essential part of the delivery mechanisms for the Long Term Plan. They are also potentially crucial in the ongoing quest to stabilise general practice for the future. Evidence from previous collaborative ventures in primary care highlights an important issue: the optimum approach to establishing collaborations and the mechanisms used to support them depend crucially on the objectives being pursued. Thus, for example, a network with an educational objective requires a different structure to one by which incentive or other payments are to be distributed. It is therefore vital that we understand at this early stage what the underpinning and animating objectives for PCNs are considered to be by those that are driving, or particularly connected to the development of, the policy, as this will affect the rules, funding mechanisms and support that need to be put in place. While the perspectives of policy makers are likely to differ from those more closely involved with front-line general practice, their perspectives are important because they determine the formal policy mechanisms which govern PCN operation.

In this paper we describe findings from the first phase of a longitudinal mixed-methods project tracking and exploring the development of PCNs and their associated outcomes. This phase of the project sought to understand in detail the national policy objectives associated with PCNs and the mechanisms by which these objectives are expected to be achieved ('programme theories'[18]). We show that the policy is underpinned by three broad groups of objectives. We then consider these alongside the content of the draft service specifications for PCNs issued late December 2019,[19] and note the revised specifications introduced as part of the updated GP contract published

in February 2020.[20] Our discussion explores the potential differences and synergies between the different objectives and highlights issues which may need to be resolved.

## PCNs: framework and funding

Forming or joining a PCN is voluntary, but practices are encouraged to engage by the provision of additional resources. The formal mechanisms by which practices work together are not prescribed, with guidance setting out a range of possible operating models with various implications associated with each.[21] PCNs have been formed as a Directed Enhanced Service (DES), that is, a nationally developed service and contractual addition to the core General Medical Services (GMS)/Alternative Provider Medical Services (APMS)/Personal Medical Services (PMS) primary care contract. The DES specification[22] requires that PCNs cover a population of 30,000-50,000 (with some flexibilities) and be geographically contiguous.

Groups of practices were invited to apply to their local commissioner (Clinical Commissioning Group; CCG) to become a PCN from 1 July 2019. As part of the registration process PCN member practices signed a 'network agreement'[23] outlining governance arrangements, including the membership list, collective rights and obligations, and financial entitlements. Additional details regarding how practices could work with each other and with other organisations are set out in seven schedules attached to the agreement, although these were not required for initial registration. Each PCN identified a local clinician to be the clinical director (CD), which guidance[22] suggested would usually be a GP. CDs are required to work collaboratively with other CDs within the integrated care system (ICS)/sustainability and transformation partnership (STP) area (interorganisational partnerships between local councils and NHS organisations working to improve care across a system) and lead engagement with other local providers.[22]

Once PCNs were approved by their CCG, they were eligible for the initial financial entitlements, which included: £1.50 per patient for participation, funded from the CCG core allocation; 0.25 full-time equivalent per 50 000 population funding for the CD; and funding for additional workforce roles. The latter was weighted according to the Carr Hill Formula, which takes some account of deprivation and burden of morbidity. The initial focus was on recruiting social prescribing link workers (funded at 100%) and clinical pharmacists (initially funded at 70%). PCNs also took over the responsibility of providing extended access routine appointments during evenings and weekends; these were previously funded by a stand-alone DES payment to practices. There was no additional funding earmarked for administrative or management costs.

The Network Contract DES will be renewed annually until 2024. The first year of the DES was framed as a development year for PCNs, with the majority of service deliverables being monitored from 2020 onwards through

additional service specifications.[21] These were described in guidance as a key component of the DES and integral to supporting the delivery of the NHS Long Term Plan.[22] The first five specifications were intended to go live in April 2020: 'Structured Medication Reviews and Medicines Optimisation', 'Enhanced Health in Care Homes', 'Anticipatory Care', 'Supporting Early Cancer Diagnosis', and 'Personalised Care'. The final two specifications, 'Cardiovascular Disease Prevention and Diagnosis' and 'Tackling Neighbourhood Inequalities', were to be implemented from April 2021. There was little detail in the initial guidance of what the programmes of work would include, with details to be agreed as part of annual contract negotiations between the British Medical Association (BMA) and NHS England. Draft service specifications were published on 23 December 2019 alongside an engagement process which ended mid-January 2020. The initial draft was not well received by the GP profession at large,[24] and in subsequent negotiations with the BMA significant concessions were made.[20] These included: increasing funding for additional roles from 70% to 100%; increasing the range of roles eligible for reimbursement; reducing the number of service specifications to be delivered in the first year; and reducing the requirements for those specifications significantly in order to reduce the associated workload. It was suggested that, with the increase in funding for staff from 70% to 100%, the participation payment of £1.50 per patient would be released to fund administrative and managerial support. Further funding was also announced for a quality incentive scheme. A number of concerns about the policy have been expressed.[25][26] In particular, commentators have highlighted the very broad range of activity that PCNs are expected to engage in and the lack of evidence underpinning some of this activity. Refining and improving the policy over time will require constructive engagement with the official objectives underlying the particular framework which has been established. This study offers the first empirical evidence of what those objectives are.

## METHODS

This paper presents findings from Work Package 1 of a longitudinal project exploring the development, operation and outcomes associated with PCNs. Three other work packages are ongoing or planned: a telephone survey of CCG PCN leads; indepth qualitative PCN case studies; and a quantitative evaluation of PCN characteristics, activities and outcomes.

### Patient and public involvement

Due to the focus of this study, no patients were involved. The broader research project, within which this study sits, will explore the extent of patient and public involvement in PCN development and activity in a subsequent work package involving PCN case studies. A patient representative sits on the project advisory group and will continue to provide feedback to inform the development of the project.

### Data collection, sampling and recruitment

Interviewees were purposively sampled for their knowledge about or role in the development of PCN policy. We sought participation from a range of relevant stakeholder organisations, including NHS England and NHS Improvement, Department of Health and Social Care, and GP representative organisations. We undertook 16 semistructured interviews with policy makers and stakeholders (July 2019–October 2019) by phone or face-to-face at interviewees' places of work. Interviews lasted between 30 and 60 min. The topic guide explored their interpretations of the objectives and associated mechanisms of the PCN policy. There was flexibility to explore interviewees' particular knowledge relating to their position or experience. Each interview was conducted by one of two experienced qualitative researchers (JH, KC), audio-recorded and professionally transcribed.

We also undertook a documentary analysis of a document published by the authority responsible for overseeing PCNs, NHS England, in December 2019 known as the 'draft service specifications'.[8] This document set out in detail proposals for the work to be undertaken by PCNs. These draft proposals can be seen as an expression of the intended outcomes that senior policy makers wished PCNs to achieve, and our analysis involved testing the document against the themes derived from our interviews in order to better understand national policy objectives and their fit with those held by other stakeholders.

### Data analysis

Transcripts were imported into NVivo V.12 (QSR International) and analysed thematically. This process involved data familiarisation, open coding, theme identification and review of themes. Coding and initial theme development were conducted by JH, but the data and themes were reviewed by the broader analysis team (JH, KC, LW-G) and refined iteratively over several rounds of analysis. The data constituting the themes were then additionally organised into a matrix by JH. Extracts were coloured by interviewee, similar extracts were grouped and connections to related extracts (mechanisms linked to particular outcomes, for example) visualised. This was used to aid decisions about what content relating to each theme should be included and emphasised in the findings, with more frequently expressed perspectives featuring most prominently. Data collection and analysis took place in parallel and the thematic matrix demonstrated clearly that data saturation was reached during our final interviews. Themes were then applied in the analysis of the draft service specifications document.[8]

## FINDINGS

We asked interviewees to explain what they thought the national policy objectives underlying PCNs were and to

describe potential or intended mechanisms. Interviewees were thus presenting their interpretations of policy objectives, not their personal beliefs about what might actually happen. We recognise that national policy makers and stakeholders may have particular perspectives about the state and needs of general practice that differ from those of others working in different parts of the system in different capacities. Our intention here is not to adjudicate between these perspectives, but to present the perspectives of those responsible for developing and implementing the rules and funding mechanisms which govern PCN operation.

We identified three main groups of objectives espoused by those with senior-level responsibility for implementing or shaping the policy: use inter-GP practice collaboration to support a primary care sector which is struggling; align primary care more closely with other community services, improving integration and service delivery; and provide a collective 'voice' for primary care in the wider system. We explore and illustrate each of these in turn. Interview extracts are denoted by a unique participant identification code in square brackets (eg, [N710cg]). We then consider the framing of policy objectives in the draft service specifications,[19] and note a shift in this in the revised specifications in the updated GP contract.[20]

### Theme 1: supporting general practice

Within this theme, PCNs represent a vehicle for supporting general practice to reduce some of the pressures it currently faces in terms of unmanageable workloads, and related challenges recruiting and retaining sufficient GPs and nurses. The key mechanism for realising this objective is the new staff that PCNs will recruit through the Additional Roles Reimbursement Scheme (ARRS). Once in place, these staff are expected to reduce the workload burden on GPs, increasing work satisfaction and subsequently improving GP recruitment and retention rates. The consequence of this is to 'rescue' general practice from the pressures it faces and increase its resilience.

Resilience is also highlighted as an overarching benefit of collaborative working 'at scale.' This involves protection against negative consequences of shocks (both endogenous and exogenous) by virtue of operating as part of a larger interorganisational entity. For example, one interviewee stated: "…networks provide an opportunity for greater resilience, so if a partner breaks their leg, the practice doesn't fall over" [N710cg]. In addition to protection for individual organisations, network membership was also expected to create other 'synergistic' benefits, such as new opportunities or increased efficiency, as a consequence of operating at a larger size. Examples offered to illustrate this included the ability to use clinical pharmacists across a collective footprint of networked practices when it would make little practical sense for any of those individual practices to employ a pharmacist for their patients alone, or the sharing of back office functions across a larger footprint.

It is the DES and associated financial incentives for GP practices that create the conditions for widespread PCN involvement. Respondents argued this collaboration would involve sharing of learning, data, and risk between practices, which would lead to improved interpractice communication and the building of greater trust. An associated outcome was a reduction in intra-PCN variation as optimal approaches are identified and adopted by networked GP practices. This will result in improved patient experience as healthcare services become more accessible to patients and better tailored to local patient need. There were also expectations that reductions in inequalities locally could be mirrored nationally once the service specifications were introduced and best practices became established nationwide.

### Theme 2: place-based interorganisational collaboration

While the theme in the previous section is concerned with inter-GP practice collaboration, this theme is defined by an emphasis on interorganisational collaborations between GP practices and other organisations and services in localities where PCNs are situated. One anticipated outcome is that more integrated and 'joined up' care will be delivered to patients in community settings. GP practices would forge closer connections to a range of local community resources and services, not just those directly health-related, and more effectively and consistently direct patients towards them. Consequently, healthcare utilisation in general, and secondary care demand (including emergency admissions) in particular, would be reduced.

> …aim is to bring together different providers in the primary care setting within networks, so within general practices, but also other providers and the voluntary sector and the community itself, to design and deliver services around specific needs of the community so to work in a networked way and try to achieve all the benefits that that would bring. [N800zf]

Interviewees recognised that it would be necessary to incentivise (non-GP) providers, such as community service providers, in order to facilitate their involvement in PCN activities towards fulfilling the aspirations of the policy, and this is planned through changes to, for example, the standard community services contract and pharmacy contract.

Some interviewees also suggested that PCNs were concerned with the development of an enhanced population health management approach whereby a range of health and other data relevant to local populations would be used to inform population segmentation, risk management assessments of particular groups and the creation of multidisciplinary teams. This would deliver a new depth of understanding about local demography and healthcare-related need.

### Theme 3: providing a 'voice' for primary care

This theme relates to PCNs' interaction with organisational entities in the broader system within which they are nested, and thus relates to both horizontal and vertical interactions rather than horizontal only. The Long Term Plan conceptualised the English NHS as a series of spatial tiers—neighbourhood, place, system—with PCNs operating at the neighbourhood level; CCGs, local councils and hospitals at the place level; and ICSs/STPs at the system level. Interviewees framed PCNs as foundational building blocks for this spatial model, integral to supporting the levels above in their operation, or as an animating force that would bring life to arrangements. More specifically, PCN CD involvement at the ICS/STP board level was highlighted as providing a means for PCNs to shape the development of the system of which they are a part and influence provider organisations at 'higher' levels. In doing so, CDs would provide a voice for primary care at the system level and represent the interests of general practice and their PCN. This is made practically more feasible by the 'at scale' approach to general practice organisation associated with PCNs: "Having a stronger voice perhaps for general practice around those particular tables that hasn't always been possible or practical to do with practices working individually" [n210×8].

### Summary

It is important to note that interviewees did not consider the objectives and mechanisms associated with each theme to necessarily be discrete or mutually exclusive. The majority primarily emphasised theme 1 'Supporting general practice' and theme 2 'Place-based interorganisational collaboration' to a lesser degree, and theme 3 'Providing a 'voice' for primary care' to a lesser degree still. One interviewee emphasised themes 1 and 3 largely equally but not theme 2. Four of the 16 interviewees gave similar weighting to the importance of all three themes. Interviewees from a background close to general practice were more likely to emphasise theme 1. Themes were also, in some cases, considered to form a temporal sequence. Some interviewees talked about 'Supporting general practice' being the necessary first step before 'Place-based interorganisational collaboration' could be more fully realised. However, others suggested it was only by GP practices working more closely with community service providers and third sector organisations that conditions in primary care would change to allow the workforce crisis to be addressed. Overall, while we have grouped the policy objectives into three overarching themes, it is clear that each was very broad, encompassing a significant number of potential objectives, mechanisms and expected outcomes. Our interviewees differed in how they envisaged the temporal sequencing of the desired objectives and in the emphasis they placed on the different groups.

### Draft service specifications

The draft service specifications were published on 23 December 2019.[19] While the introduction to the document references reducing GP workload[19] (p4), the focus within the specifications is on the delivery of additional services by PCNs. Two of the five services (Structured Medication Reviews and Medicines Optimisation, and Enhanced Health in Care Homes) were intended to be fully implemented from April 2020, with three more—anticipatory care, personalised care and supporting early cancer diagnosis—introduced from April 2020 in a phased manner over successive years until 2023/2024 in order to avoid '…overburdening [PCNs] at an early stage with unrealistic expectations for new service delivery' (p4).[19]

The specification document offers a clear programme theory for the policy:

> Through a combination of the additional workforce capacity within primary care, and the redesign of community services provision to link with and support PCNs, we expect the Network Contract DES both to reduce workload pressures on GPs and support improved primary care services to patients.[19] (p4)

It is also suggested that the additional workforce recruited will be sufficient to cover all work associated with delivering the five service specifications, while simultaneously providing spare capacity to take up some work currently undertaken by GPs. Thus, it is claimed the workload burden on practices will reduce, although no evidence is provided to support this.

Structured medication reviews are to be delivered by individual practices, supported by clinical pharmacists. However, to perform these checks pharmacists will need prescribing qualifications, and not all pharmacists being recruited have this extra training. This work is therefore likely to devolve to GPs and any nurses with prescribing qualifications. Enhanced care in care homes will be delivered in collaboration with community service providers, as will anticipatory care, both of which require the establishment of network-level multidisciplinary teams. The personalised care service specification references better linkage with voluntary community groups, alongside the provision of personal health budgets. It is suggested this will enhance population health and reduce secondary care service use. Finally, the supporting earlier cancer diagnosis specification references greater collaboration between GPs and other service providers such as cancer alliances, secondary care and public health teams. The document explicitly references the intention that delivering the service specifications will lead to greater cooperation between GPs and community services, and suggests that this will be enhanced by forthcoming changes to the standard community services contract.

Taking the draft service specifications as a whole, the intention that PCNs will support the greater integration between primary care and community and other services comes through as the strongest underlying policy

objective. References to practice workload are present, but only in so far as to make the argument that delivering these service specifications will have a beneficial effect on that workload, thereby supporting general practice. Little concrete evidence is provided to support these arguments, beyond some general statements of mechanisms by which the services are expected to improve patients' health and therefore reduce demand.

The poor response from the profession to this document resulted in substantial changes during negotiations between the BMA and NHS England, including significantly increased funding, reduced requirements associated with the service specifications and increased flexibility in the ARRS.[20] The document setting out the revised deal also offers a shift in tone towards our first theme, with a greater emphasis on reducing workload for GPs.[20]

## DISCUSSION

Previous studies of GPs working together in a variety of ways have shown that collaborations can be associated with beneficial outcomes and that job satisfaction may increase.[13 16] However, the exact characteristics of particular collaborative ventures can have an impact on participation.[26] The design of the PCN scheme and its embedded incentives therefore matters a great deal, and the policy context surrounding PCNs is moving rapidly. Our interviews illuminated a broad range of policy objectives to be achieved by PCNs, with three themes dominating the accounts: the need to support general practices; the desire to improve collaborative working between GP practices and other community-based services; and providing a collective voice for primary care within an evolving, more integrated, system. Importantly, we found our interviewees varied in terms of the emphasis they placed on different groups of objectives, with those most closely linked to primary care tending to stress the need to support practices, while those more distant from practice more likely to focus on the development of a more integrated system of community-based services. The objectives are clearly not mutually exclusive but may require temporal sequencing, with the stabilisation of a struggling primary care sector probably needing to occur before meaningful engagement with other community service providers can be achieved and before a 'collective voice' can be agreed.[25 26] Moreover, different objectives imply different approaches to prioritising investment and may necessitate different incentives,[27 28] while approaches to providing support and training may differ according to which objectives are being pursued. For example, CDs will require different knowledge and skills to engage with ICS than to work across practices to broker trust and engagement internally.[29 30]

Against this backdrop it is somewhat unsurprising that the draft service specifications released in December 2019 were not well received, as these almost exclusively focused on the delivery of Long Term Plan objectives around joined-up community-based services, with references to practice workload limited to arguments as to why delivery of these services will not increase workload. For those for whom the prime early objective of PCNs is to support general practice, such arguments are unlikely to be convincing. In subsequent negotiations the draft was changed significantly with greater emphasis on practice workload and significantly increased funding.[20] The manner in which this framing endures and evolves over time is likely to have a bearing on the success of the implementation of the policy and warrants continued attention.

## Strengths and limitations

This is an early study focusing on policy objectives as perceived by those in senior positions across the system. As such, it inevitably offers a snapshot of a rapidly developing situation. Our wider study is ongoing and will explore the development of PCNs over time in more depth, as well as examining outcomes. We interviewed a relatively small number of people; however, participants were purposively selected to enable us to explore the interface between policy design and implementation as events unfolded, from the perspectives of some of those most closely involved.

## CONCLUSION

By exploring how policy makers and key stakeholders conceptualise the objectives of the PCN policy, and the variation between them, this study offers an explanation for the problems associated with the initial draft of the service specifications, going beyond a simple explanation based around workload to demonstrate a potential mismatch in objectives for the policy between different stakeholder groups. This offers policy makers some suggestions as to how confidence in the PCN policy might be restored in the longer term. All those we spoke to emphasised the general enthusiasm for collaboration between practices, and as we have highlighted such collaborations have a long history. Our findings suggest that early focus on measures to support practices may be rewarded by opportunities to deliver the broader integration agenda in the longer term. Slowing down the implementation of the policy may also be helpful.

More generally, our study highlights the importance of understanding the programme theories and objectives underlying policies. The PCN policy is wide-ranging, and the successful establishment of PCNs may be able to offer multiple benefits to the health system. However, a failure to clarify the objectives behind the policy and a mismatch between the objectives of different groups may have played a part in the difficulties associated with the draft service specifications. Policy ambiguity has long been used by policy makers to enable the development of implementation coalitions,[31 32] but such ambiguity cannot be maintained in the face of concrete requirements for delivery. Furthermore, choosing a contractual mechanism for implementation (rather than, eg, a piloting approach

using incentives) necessarily constrains local implementation discretion and places limits on what can be done. A national contract with fixed funding models and standard delivery requirements implies certainty about what activity is required, something which is rarely present in health and care contracting.[33 34] Such a contract limits opportunities for PCNs to respond to local contextual conditions or to choose their own priorities, and this may be problematic in the longer term. At the very least, our study confirms the need for those responsible for setting contract requirements to understand the range of objectives espoused by those who must act to implement the policy, and suggests that delivery requirements should be crafted in such a way that all groups can see some opportunity to meet their objectives with flexibility to allow adaptation to local needs.[35]

**Acknowledgements** The authors wish to thank the interviewees for their time and contribution to the study, and the broader PRUComm team.

**Contributors** KC designed the study. JH and KC conducted the interviews relating to this paper. JH led the analysis of the data with support from KC, LW-G and SB. KC and JH drafted the manuscript, to which all authors made substantial contributions. All authors approved the final version and agree to be accountable for all aspects of the analysis.

**Funding** This research is part of independent research commissioned and funded by the NIHR Policy Research Programme (ref: PR-PRU-1217-20801).

**Disclaimer** The views expressed in this publication are those of the authors and do not necessarily reflect those of the Policy Research Programme, NIHR, NHS England, the Department of Health, arm's length bodies or other government departments.

**Competing interests** None declared.

**Patient consent for publication** Not required.

**Ethics approval** This study was granted ethical approval by The University of Manchester Proportionate Research Ethics Committee (study number: 2019-6922-11622). Participants were provided written information about the study. They provided written consent or gave consent verbally at the beginning of telephone interviews.

**Provenance and peer review** Not commissioned; externally peer reviewed.

**Data availability statement** No additional data are available.

**ORCID iD**
Jonathan Hammond http://orcid.org/0000-0002-4682-9514

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
