## [Reviewer comments · BMJ Open]

ARTICLE DETAILS

TITLE (PROVISIONAL)	Exploring the multiple policy objectives for Primary Care Networks: a qualitative interview study with national policy stakeholders
AUTHORS	Checkland, Kath; Hammond, Jonathan; Warwick-Giles, Lynsey; Bailey, Simon

VERSION 1 – REVIEW

REVIEWER	Tim Wilson Oxford Centre for Triple Value The Oxford Centre for Triple Value Healthcare is a social enterprise that receives fees for advising health services on policy issues
REVIEW RETURNED	23-Mar-2020

GENERAL COMMENTS	Overall I thought this paper useful and worthy of publication. However, there are some areas that would require revision before I would support publication. The authors should acknowledge that national policymakers and stakeholders, interviewed for this project, will have a somewhat distorted view of the state and needs of general practice. This lack of understanding is demonstrated by the second rejection of the PCN contract negotiated by the very same people the authors interviewed. And the same people that have been in positions of leadership leading to "Primary Care in the UK is in crisis" as the authors put it. This does not invalidate the paper, but needs to be referenced throughout (not just at the end of the paper) as a limitation, or a factor in their interpretation. The authors do not reference important recent BMJ articles, including by Richard Murray, Stephanie Kumpunen and Richard Lewis and myself. Richard Murray's editorial is especially important as it draws exactly the same conclusions that the authors do. The abstract opens using the word "voluntary", but this is in contrast to the introduction of the paper which sets the context better. Perhaps the opening abstract statement should be: "During a time of crisis, UK general practice is being offered extra funding but only if they participate in collaborations between general practices." In their conclusions, the authors might have pointed out the highly transactional nature of the relationship between NHS England and general practice. For a complex service like general practice, transactional contracting is bound to fail (see Williamson, Hart, Le Grand etc.).
---

	Minor points P4 line 23-24. The reasons for GPs leaving are more complex and are addressed in the review they cite, but not included here. P10 line 16-17 I am surprised that shared "back office" functions and scale did not come up as a policy objective. I have certainly heard this discussed by senior officials.
--	---

REVIEWER	Bernard Le Floch Université de Bretagne Occidentale , General Practice EA 7479 SPURBO
REVIEW RETURNED	07-Apr-2020

GENERAL COMMENTS	Bernard Le Floch Université de Bretagne Occidentale , General Practice EA 7479 SPURBO 07-Apr-2020 Exploring the multiple policy objectives for Primary Care Networks: a qualitative interview study with national policy stakeholders Thank you to permit me to review this article. The study's aim was to explore national policy objectives underpinning PCNs. Introduction: P4 L23: Introduction: Primary care in the UK is in crisis. I understand the objectives are to recruit GPs. This article aimed to research the positive aspects of NCPs seen not from the side of doctors, but from the side of decision-makers. This research may seem strange. How policymakers decide what improves job satisfaction for doctors? But the study can be interesting. One of the difficulties I had to read your article is that as French, I do not know much about the principle of NCPs. Even if multidisciplinary groups exist with us. It would be good to clarify the spirit of the project. Method: A more detailed description of the research steps could be useful in understanding the interpretation of the results. I do not understand where is the "policy document analysis". I did not found a table with the description of the participants. How many GPs? Are they working in a practice? L 40: "We undertook 16 semi-structured interviews with policy makers and stakeholders (07/2019-10/2019) by phone or face-to-face": This are results for me. Was saturation reached? And how? I understood that one of the motivations for supporting NCPs is the satisfaction of GPs found in inter-professional collaboration (theme 1). This factor of job satisfaction is known: " Le Floch B, et al. Which positive factors give general practitioners job satisfaction and make general practice a rewarding career? A European multicentric qualitative research by the European general practice research network. BMC Fam. Pract. 2019;20:96." In this article, one of the main factors was the patient. I'm surprised it doesn't show up in your results. Maybe it comes from the participants? And what about relationships with the other care providers? And about personal life: family, leisure? ...
---

	P8, L15: "... to design and deliver services around specific needs of the community so to work in a networked way and try to achieve all the benefits that that would bring." [N800zf] There are not a lot of quotes in this article. For this one, did the participant explained which "benefits" he wanted to say? Conclusion: I have difficulties to see how your conclusion is derived from your finding. The first lines seem to be discussion, with references. My conclusion after reviewing this article: For me, this article is interesting and can be published. But it is necessary to make some improvements because it must be readable by an international audience. The method needs to be better explained. Recommendation to the editor: This manuscript is not adequate for publishing as it is presented now. It may be re-considered after revisions.
--	--

VERSION 1 – AUTHOR RESPONSE

	Comment (R1)	Response
1	The authors should acknowledge that national policymakers and stakeholders, interviewed for this project, will have a somewhat distorted view of the state and needs of general practice. This lack of understanding is demonstrated by the second rejection of the PCN contract negotiated by the very same people the authors interviewed. And the same people that have been in positions of leadership leading to "Primary Care in the UK is in crisis" as the authors put it. This does not invalidate the paper, but needs to be referenced throughout (not just at the end of the paper) as a limitation, or a factor in their interpretation.	Thank you for highlighting this. We agree that the perspectives that policy makers hold on general practice is an important factor in contextualising our findings, although we would argue that the extent to which such views are distorted or differ from those working in general practice is an empirical question. To make this recognition more explicit, we have inserted a sentence in the introduction which provides a more explicit justification for our focus on policy makers' perspectives. We have highlighted the fact that, whether policy makers' perspectives are 'distorted' or not, they will determine the formal rules and funding mechanisms put in place to govern PCNs. Given that the success or otherwise of PCNs in the long term will be affected by these rules, we feel that it is very important to understand the attitudes and beliefs that have underpinned their development and governance. We hope that this is now clearer from our introduction. In addition, we have inserted the following text into the first paragraph of the findings: "We recognise that national policy makers and stakeholders may have particular perspectives about the state and needs of general practice that differ from those of others working in different parts of the system in different capacities. Our intention here is not to adjudicate between these

		perspectives, but to present the perspectives of those responsible for developing and implementing the rules and funding mechanisms which govern PCN operation.”
2	The authors do not reference important recent BMJ articles, including by Richard Murray, Stephanie Kumpunen and Richard Lewis and myself. Richard Murray's editorial is especially important as it draws exactly the same conclusions that the authors do.	This is a fast moving policy area, and we are grateful to the reviewer for highlighting relevant commentaries which we may have missed. We have added a reference to these in the section describing the policy, and explained more explicitly why we think that mitigating the concerns expressed by those with knowledge of primary care will require engagement with the policy objectives held by those with control over the structures and rules put in place. We have made more explicit the fact that our study offers the first empirical evidence about these issues.
3	The abstract opens using the word "voluntary", but this is in contrast to the introduction of the paper which sets the context better. Perhaps the opening abstract statement should be: "During a time of crisis, UK general practice is being offered extra funding but only if they participate in collaborations between general practices."	We are grateful for this suggested change to the framing of the paper in the abstract and have used it as a basis for a change to the wording.
4	In their conclusions, the authors might have pointed out the highly transactional nature of the relationship between NHS England and general practice. For a complex service like general practice, transactional contracting is bound to fail (see Williamson, Hart, Le Grand etc.).	Thank you for drawing our attention to this. We agree, and have reflected this point in the discussion, highlighting the lack of flexibility associated with a transactional contraction approach and considering how this might be mitigated.
5	P4 line 23-24. The reasons for GPs leaving are more complex and are addressed in the review they cite, but not included here.	Reworded to read: "Primary care in the UK is in crisis. Young doctors are not entering, or remaining in, the speciality in sufficient numbers to cope with demand, many older GPs are increasingly dissatisfied with general practice due to various intrinsic and extrinsic factors and choosing to retire early (1, 2)."
6	P10 line 16-17 I am surprised that shared "back office" functions and scale did not come up as a policy objective. I have certainly heard this discussed by senior officials.	Sharing back office functions and increased scale fall under theme 1 'supporting general practice.' The relevant policy objective highlighted by interviewees was of increased resilience for general practice as a result of operating at a larger scale and the increases in efficiency that this could

		provide. We have added a reference to this to make it clearer in the text
	Comment (R2)	Response
1	P4 L23: Introduction: Primary care in the UK is in crisis. I understand the objectives are to recruit GPs. This article aimed to research the positive aspects of NCPs seen not from the side of doctors, but from the side of decision-makers. This research may seem strange. How policymakers decide what improves job satisfaction for doctors? But the study can be interesting.	We are glad that the reviewer sees some interest in our approach. Our argument is that, in the context of a workforce crisis, policymakers are implementing wide ranging policy solutions designed to relieve the crisis. Understanding what policy makers are trying to achieve, and how they think that their policy solution will contribute to achieving their objectives is a vital step in judging the success of the policy as well as in understanding how it is implemented and operationalised. Primary Care Networks are an interesting policy because they are being heralded as the solution to many different problems. How they are designed will be crucial in determining which of the multiple objectives can be achieved.
2	One of the difficulties I had to read your article is that as French, I do not know much about the principle of NCPs. Even if multidisciplinary groups exist with us. It would be good to clarify the spirit of the project.	We highlight the explicit purpose of the PCN policy in the first paragraph of the introduction, and have attempt to clarify this further by amending it to read: “The underlying idea is relatively simple: incentivise GP practices to combine together into groups, so they can find economies of scale, employ a wider range of staff, link more effectively with community-based providers and, through this, improve services to patients.”
3	Method: A more detailed description of the research steps could be useful in understanding the interpretation of the results. I do not understand where is the “policy document analysis”. I did not found a table with the description of the participants. How many GPs? Are they working in a practice? L 40: “We undertook 16 semi-structured interviews with policy makers and stakeholders (07/2019-10/2019) by phone or face-to-face”: This are results for me.	The policy document analysis in the paper relates to the PCN draft service specifications. This is the final sub-section in the Findings. We have not included a table or any additional detail regarding the interviewees because the pool of potential participants is relatively small and we wish to ensure we preserve the anonymity of those that took part. Interviewees were recruited on the basis of their role as policy makers or national level stakeholders. We have not specified how many were GPs in addition to their policy-related role, again, to preserve anonymity.

	Was saturation reached? And how?	Thank you for your point regarding saturation, we have added the following comment in the data analysis section to clarify this: "Data collection and analysis took place in parallel and the thematic matrix demonstrated clearly that data saturation was reached during our final interviews."
4	I understood that one of the motivations for supporting NCPs is the satisfaction of GPs found in inter-professional collaboration (theme 1). This factor of job satisfaction is known: " Le Floch B, et al. Which positive factors give general practitioners job satisfaction and make general practice a rewarding career? A European multicentric qualitative research by the European general practice research network. BMC Fam. Pract. 2019;20:96."	Our interviewees emphasised that the additional roles recruited into general practice through PCNs would improve GP recruitment and retention primarily by reducing the workload pressures GPs faced, rather than by improving their job satisfaction as a result of increased inter-professional collaboration. In this paper we are not attempting to find evidence for or against particular assertions of the policy; we are interested in highlighting the understanding that policy makers have about the objectives of the policy and how these might be realised.
5	In this article, one of the main factors was the patient. I'm surprised it doesn't show up in your results. Maybe it comes from the participants? And what about relationships with the other care providers? And about personal life: family, leisure? ... P8, L15: "... to design and deliver services around specific needs of the community so to work in a networked way and try to achieve all the benefits that that would bring."[N800zf] There are not a lot of quotes in this article. For this one, did the participant explained which "benefits" he wanted to say?	Potential benefits to patients from PCNs features in both themes 1 and 2. When explaining their understanding of the policy's aims, however, interviewees did not usually emphasise improvements in patient experience and outcomes, and those that did tended not to specify a clear mechanism for how PCNs would realise this. The extent and manner of attention paid to patients in the results reflects this. The interviewee was rather non-specific about the benefits of network working beyond it offering the potential to make service provision more closely tailored to the particular needs of local populations. We have included only a relatively small number of interview extracts because of the constraints of the word limit.
6	Conclusion: I have difficulties to see how your conclusion is derived from your finding. The first lines seem to be discussion, with references.	Thank you for this suggestion. In the interests of making the conclusion more focussed, we have moved the first section of the conclusion to the beginning of the discussion.

VERSION 2 – REVIEW

REVIEWER	Dr Tim Wilson Oxford Centre for Tripe Value Healthcare The Oxford Centre for Tripe Value Healthcare is a small social enterprise that receives fees for supporting health services build capacity and capability to move to value-based healthcare
REVIEW RETURNED	27-Apr-2020
GENERAL COMMENTS	I believe the authors have fully addressed the issues raised in the previous draft.
REVIEWER	LE FLOCH Bernard Université Brest Occidentale
REVIEW RETURNED	02-May-2020
GENERAL COMMENTS	Thank you for the revised manuscript. The paper improved considerably, especially in terms of clarity of the topics, the aims and the methodological dispositions. I think this is an original study and I enjoyed reading and reviewing it. Prof Bernard Le Floch MD, PhD.